# The Role of NK Cells in EBV Infection and EBV-Associated NPC

**DOI:** 10.3390/v13020300

**Published:** 2021-02-15

**Authors:** Yi Tian Png, Audrey Zhi Yi Yang, Mei Ying Lee, Magdalene Jahn May Chua, Chwee Ming Lim

**Affiliations:** 1Department of Otorhinolaryngology–Head and Neck Surgery, Singapore General Hospital, Singapore 169856, Singapore; png.yi.tian@sgh.com.sg (Y.T.P.); audreyyang97@gmail.com (A.Z.Y.Y.); lee.mei.ying@sgh.com.sg (M.Y.L.); magdalene.chua.j.m@sgh.com.sg (M.J.M.C.); 2Surgery Academic Clinical Program, Duke-National University of Singapore Medical School, Singapore 169856, Singapore

**Keywords:** NK cells, Epstein-Barr Virus (EBV), Nasopharyngeal Carcinoma (NPC)

## Abstract

A vast majority of the population worldwide are asymptomatic carriers of Epstein-Barr Virus (EBV). However, some infected individuals eventually develop EBV-related cancers, including Nasopharyngeal Carcinoma (NPC). NPC is one of the most common EBV-associated epithelial cancers, and is highly prevalent in Southern China and Southeast Asia. While NPC is highly sensitive to radiotherapy and chemotherapy, there is a lack of effective and durable treatment among the 15%–30% of patients who subsequently develop recurrent disease. Natural Killer (NK) cells are natural immune lymphocytes that are innately primed against virus-infected cells and nascent aberrant transformed cells. As EBV is found in both virally infected and cancer cells, it is of interest to examine the NK cells’ role in both EBV infection and EBV-associated NPC. Herein, we review the current understanding of how EBV-infected cells are cleared by NK cells, and how EBV can evade NK cell-mediated elimination in the context of type II latency in NPC. Next, we summarize the current literature about NPC and NK cell biology. Finally, we discuss the translational potential of NK cells in NPC. This information will deepen our understanding of host immune interactions with EBV-associated NPC and facilitate development of more effective NK-mediated therapies for NPC treatment.

## 1. Introduction

Epstein-Barr Virus (EBV), also known as human gammaherpesvirus 4 (HSV-4), is commonly spread through saliva and carried by a majority of the world’s population. A vast majority of the infected humans are asymptomatic; the virus establishes latency in resting B memory cells in these lifelong carriers [1,2]. Individuals can, however, develop infectious mononucleosis (IM), an acute symptomatic condition when the virus is acquired late in childhood [3,4]. Despite its quiescent nature in most individuals, EBV is strongly associated with several B cell and epithelial malignancies [5,6]. Epithelial malignancies account for nearly 80% of all EBV-associated cancers, with Nasopharyngeal Carcinoma (NPC) being one of the most common EBV-associated epithelial cancers, which carries an annual prevalence of 78,100 diagnosed cases reported worldwide [7].

EBV-associated NPC is a common head and neck cancer highly endemic in Southern China and Southeast Asia [8]. Besides EBV infection, established risk factors of NPC include Cantonese ethnicity, male gender, family history of NPC, high consumption of preserved foods and several human leukocyte antigen (HLA) class I alleles [9]. Common clinical presentations of NPC include a mass in the neck, blood-stained nasal discharge, hearing loss and cranial nerve palsies [10]. Because of its “hidden” location and non-specific symptoms, NPC are typically diagnosed in advanced stages (III–IV), resulting in poorer prognosis. The five-year survival rate for patients in stages I, II, III and IV are 66.7%, 55.6%, 41.8% and 25.9%, respectively [11]. For primary NPC, radiotherapy is the mainstay of treatment, with concurrent chemoradiotherapy prescribed for locally advanced cases. Although NPC is highly sensitive to radiotherapy and chemotherapy, around 15% to 58% of patients experience recurrent disease, for which there are currently no long-term or durable treatment options available for these patients with recurrent cancer [12]. Surgical salvage can treat approximately a third of patients with locoregional recurrence; even so, distant recurrence following successful locoregional control remains high at 30–40% [13,14]. Furthermore, there is a 30% risk of grade 4–5 toxicities (including treatment-related deaths) among patients who receive re-irradiation for locoregional recurrences [15]. Patients with distant recurrences are treated with palliative chemotherapy with an overall survival and progression-free survival of 12–15 months and 3–5 months, respectively [13,16,17].

Because of the strong immune responses against EBV, NPC is characterized by dense lymphocytic infiltrates in the tumor stroma [18]. Studies on immune infiltrates in NPC reveal select immune phenotype compositions that may affect disease prognosis [19,20]. The presence of EBV-specific cytotoxic T cells in NPC demonstrates that these EBV epitopes may be targeted for immunotherapy [5,21,22]. Refractory NPC patients have been treated with infusion of EBV-primed cytotoxic T lymphocytes (CTLs) with some success [5,23,24,25], validating the role of cell therapy in NPC. The review of this treatment in NPC was reported in our previous publication [26]. Nonetheless, downregulation of Human Leukocyte Antigen (HLA) class I, a molecule required for priming and activation of CTLs, are common tumor evasion mechanisms among EBV-associated NPC [27,28,29]. This loss of HLA class I molecules on NPC, however, presents an opportunity for Natural killer (NK) cells to kill these cancer cells, since NK cell-mediated cytotoxicity is not HLA dependent. Therefore, understanding the interactions between NK cells, EBV and NPC will aid in shaping novel therapeutic interventions in EBV-associated NPC.

## 2. EBV Biology and Its Role in NPC

EBV is a dual-tropic virus that infects both B lymphocytes and epithelial cells. Infection of both cell types in the pharynx in areas such as the tonsils have been shown to be crucial for viral persistence in the host [30,31]. The virus can switch between lytic and latent phases; the former facilitates propagation and spreading of virus progenies, while the latter avoids immune detection and elimination by its host.

EBV infection transforms B cells to proliferative lymphoblastoid cell lines (LCLs) in vitro, resembling plasmablasts or early plasma cells found in germinal centers [4,32,33]. EBV-transformed LCL-like cells are observed particularly when the host’s immunity is compromised, and are seen in conditions such as post-transplant lymphoproliferative disorder (PTLD) or AIDs [34]. This suggests that the in vivo persistence of these EBV-infected cells is mediated by the immune system. Transition to type 0 or I latency (i.e., absence or limited expression of only viral products Epstein-Barr nuclear antigen 1 (EBNA1) and Epstein-Barr virus-encoded small RNAs (EBER), respectively) during differentiation of infected B cells to resting B memory cells is also crucial for long-term persistence of the virus, as viral replication is induced only after memory cell differentiation to plasma cells [35].

Establishment of EBV latent infection in epithelial cells is widely agreed to be an early oncogenic event preceding NPC. This is supported by how latent infection is only detectable in precancerous lesions and invasive NPC, but not normal nasopharyngeal epithelium [36,37,38,39]. Nonetheless, EBV infection of normal epithelial cells do not induce LCL-like clonal proliferation [40,41,42]. Rather, it results in production of high levels of virions, causing cytopathic changes common in lytic virus phase without alterations to host cell proliferation and differentiation [38,43]. Furthermore, induction of EBV latent gene expression is only feasible in immortalized epithelial cells with altered gene expression that overcome cell cycle arrest (e.g., cyclin-D overexpression or p16 inactivation) or in presence of inflammatory cytokine TGF-β1 [42,44]. Strikingly, cyclin-D overexpression and/or p16 inactivation is commonly observed in premalignant nasopharyngeal epithelium [9,44,45]. These suggest that other factors, such as the host genotype and acquired genomic/epigenetic changes, are required for malignant transformation of the EBV-infected epithelium.

In EBV-NPC, tumor cells are known to undergo lytic reactivation [46,47], evidenced by the high EBV DNA load and high levels of anti-EBV viral capsid antigen (VCA) antibodies in both serum [48,49] and nasopharyngeal swab samples [50] of NPC patients. More evidence further suggests the contribution of lytic reactivation to NPC progression, such as the increased DNA double-strand breaks and micronuclei formation (marker of genome instability) [51], and the presence of EBV integration at vulnerable regions of the genome close to tumor-suppressor and inflammation-related genes [52]. Additionally, several EBV lytic proteins were noted to contribute to genome instability [53,54,55,56]. Furthermore, Zta, an EBV lytic protein, may promote angiogenesis through induced production of vascular endothelial growth factor (VEGF) [57]. Hence, while latent genes provide survival signals to NPC, lytic phase promote NPC progression through genome instability and angiogenesis.

## 3. NK Cells

NK cells (CD56^+^CD3^-^) make up 1–20% of peripheral blood mononuclear cells in humans [58,59]. They can be divided into two broad categories based on expression levels of CD56 and CD16: CD56^dim^CD16^+^ and CD56^bright^CD16^−^. About 90% of NK cells are CD56^dim^CD16^+^ (“mature” cytotoxic NK cells) and are found in the blood, bone marrow, spleen and lungs with high cytotoxic potential; they produce high levels of Interferon (IFN-γ), Tumor Necrosis Factor (TNF-α) and cytolytic mediators perforin and granzymes. The other 10% of NK cells are CD56^bright^CD16^−^ (“immature” immunomodulatory NK cells) and are found in lymphoid tissues, tonsils and gut; they have poorer cytotoxic potential, but can be stimulated to produce IFN-γ in presence of Interleukin (IL)-12 and IL-18 [60].

Similar to CTLs, NK cells target both virus-infected cells and nascent transformed cells [61,62,63]. They do so, however, through germline-encoded receptors, unlike CTLs, which rely on somatically rearranged receptors. Activation of NK cells occurs when there is more stimulation of its activating receptors than inhibitory receptors. Conversely, NK cell inhibition occurs when there is more inhibitory receptor than activating receptor stimulation. Since activated NK cells can elicit direct cell cytolysis of target cells, they are considered first-line defenders against viral and nascent aberrant cell transformation.

## 4. How Do NK Cells Process EBV Infection?

In acute phase of symptomatic EBV-associated infectious mononucleosis (IM), marked expansion of NK cells and EBV-specific T cells indicates the importance of these immune cells in controlling EBV infection [3,64]. Enriched NK population in IM patients are due to preferential expansion of CD56^dim^NKG2A^+^Killer cell Immunoglobulin-like Receptor (KIR) early-differentiated NK cells [65]. Notably, isolated NK cells from IM patients are less efficient at elimination of autologous LCL, due to the high expression of HLA class I in latently infected B cells [64,65]. Importantly, isolated NK cells from pediatrics IM patients exhibited greater effector function against EBV-infected B cell line in lytic phase [65]. Interestingly, lower frequencies of CD56^dim^NKG2A^+^KIR^-^ NK cells were observed in patients with PTLD than in IM pediatric patients, and may possibly account for suboptimal anti-EBV immune control in the former [66]. In humanized mice (HuNSG) with NK depletion, acute IM-like symptoms were aggravated when these mice were challenged with EBV [67]. Compared to EBV-challenged HuNSG mice without NK depletion, those with NK depletion had larger spleen, higher viral titers, greater weight loss and greater presence of infiltrating lymphomatous tumors [67]. These data show that NK cells are crucial in limiting lytic EBV infection before the EBV-specific adaptive response kicks in to control the disease.

It is clear that the presence of EBV-specific T cells alone is not sufficient to control EBV infection [67,68]. In total, 53% of classical natural killer cell deficiency (CNKD) patients experienced severe herpesvirus infections, with some developing severe EBV-driven diseases. Additionally, one CNKD patient had recurrent EBV-driven lymphoproliferative disease despite developing adaptive immunity against EBV [68]. EBV-associated diseases have been reported in patients with mutations that diminish/abrogate NK activity. NK cytolytic activity can be inhibited by mutations of Src homology 2 domain-containing protein (SH2D1A), which affects 2B4 signaling. NK cells isolated from patients with SH2D1A mutations were unable to kill EBV-positive B cell lines [69]. Persistent symptomatic viremia and EBV lymphoproliferative disorder was also observed in patients with a deficiency in CD27, which is a costimulatory molecule found on lymphocytes, including NK cells [70,71].

Numerous in vitro studies have established that EBV-infected B cells are more susceptible to NK-mediated lysis in lytic phase than in latent phase. This important finding was demonstrated as early as 1980 by Blazar, Patarrovo and colleagues, who showed that EBV-carrying tumor B cell lines were more susceptible to NK killing when they were induced from the latent phase into the lytic phase [72,73]. This was later supported by Pappworth and colleagues, who demonstrated how latently infected B cell line were resistant to cell-mediated lysis by NK cell line and primary polyclonal NK cells [27]. Similarly, lytic phase induction of these EBV B cell lines sensitized them to NK cell-mediated cytotoxicity. Corresponding downregulation of HLA class I, HLA-E molecules, and the upregulation of NK-activating ligands, UL16-binding protein 1 (ULBP-1) and CD122 were observed on the EBV-infected B cell lines [27]. Notably, the expression of BZLF-1 alone, a lytic phase gene, was sufficient to make the EBV-infected B cell lines vulnerable to NK killing [74]. Nonetheless, there are EBV-encoded proteins that can counteract NK-mediated lysis during late-lytic/pre-latent phase [74,75]. EBV-infected B cells with *BCRF1* gene knockout, which abrogate the production of viral homologue of IL-10 (vIL-10), have been shown to sensitize pre-latent phase EBV infected cells to NK-mediated lysis [75]. *BHRF1* gene, which encodes viral homologue of Bcl2 protein (vBcl2) during the late lytic phase, also promotes resistance of these EBV-infected B cell lines against NK-mediated killing through enhanced resistance to apoptosis [74].

NKG2A^+^ NK cells appear to be the main effectors which are most efficient at killing EBV-infected B cells. This was first suggested by Azzi and colleagues when they noted the preferential enrichment of NKG2A^+^, early-differentiated NK cells in IM patients [65]. Hatton and colleagues also demonstrated that NKG2A^+^ NK cells from healthy peripheral blood mononuclear cells (PBMC) were more efficient in killing autologous LCL compared to NKG2A^-^ NK cells [76]. Co-cultures of NKG2A^+^ NK cells with autologous LCL induces the NK cells’ production of IFN-γ and lytic degranulation [76]. Further immune phenotyping revealed NKG2A^+^2B4^+^CD16^-^CD57^-^NKG2C^-^NKG2D^+^ as the main peripheral NK subset from healthy PBMC that targets LCL [76]. Nonetheless, these studies are performed on circulating NK cells; and tissue-resident NK, especially tonsillar NK, may play a larger role in restricting EBV infection since tonsils are the direct portal of entry for EBV.

Tonsillar NK cells likely serve as the main innate barrier against EBV infection as highlighted in some studies [77,78,79]. Strowig and colleagues noted that the tonsillar NK subset, CD56^bright^CD16^-^ NK cells, responded readily to matured dendritic cells (DC) and secreted superior levels of IFNγ than peripheral blood NK cells [77]. High IFNγ concentrations, produced by tonsillar NK cells, were potent enough to reduce proliferation of EBV-infected B cells through delayed expression of latent EBV antigens [77]. The same group later identified CD56^bright^NKG2A^+^CD94^+^CD54^+^CD62L^-^ as the main CD56^bright^ tonsillar NK subset which restricted EBV-mediated B cell transformation and was found to accumulate in the tonsils of EBV-infected children compared to non-infected children and adults [78]. Additionally, these tonsillar NK cells were more mature, and expressed significantly greater KIRs and CD16 than other tonsillar CD56^bright^ NK [78]. A more recent study further confirmed that the tonsillar CD56^bright^NKG2A^+^ NK cells were the most potent subset in limiting the proliferation of autologous EBV-infected tonsillar B cells, specifically of the germinal center or naïve B cell subset [79]. These tonsillar CD56^bright^ NKG2A^+^ NK subset could be the precursors of peripheral CD56^dim^NKG2A^+^ cells found in patients with acute IM [65,79]. The authors further demonstrated these NK2GA^+^ subset control EBV infection through IFN-γ release; and partially through NKp44 engagement [79]. Taken together, these studies highlight the strong potential of tonsillar CD56^bright^NKG2A^+^ NK cells in limiting EBV-infected B cells.

Finally, NK activating receptors such as NKG2D, NKp44, CD16 and DNAM are relevant in NK-mediated restriction of EBV. The summary of these receptors and relevant cytokines in mediating EBV restriction is presented in Table 1 and Table 2, respectively.

## 5. How Does EBV Latency Contribute to NK Cell Evasion?

EBV is known to exhibit four types of latency programs: type Ⅲ, type Ⅱ, type I and type 0 latency. Each type presents different characteristics that serves the virus in the different stages of its life cycle. Among the four, the type II latency program is specifically adopted by the virus in NPC where the infected cells express a specific, limited set of viral genes, including latent phase antigens (LMP1), LMP2A/B, EBNA1, EBERs and BamHI-A rightward transcripts (BARTs), of which there has been evidence of their contribution to NK evasion [9]. Type 0, type Ⅰ and type Ⅲ latency programs are less relevant in the context of NPC, thus our focus here will be on EBV type Ⅱ latency.

When EBV-infected cells enter the latency phase, they are protected against NK cell killing. This observation was proven when a lytic phase-defective EBV strain (with *BZLF-1* knockout) was inoculated in HuNSG mice; the resulting default EBV latent infection could not induce NK cell expansion as opposed to mice challenged with wild-type EBV [67]. Restricted expression of lytic antigens can also shield infected cells from antibody-dependent mediated clearance by NK cells [80]. This contrasts with latent phase antigens (LMP1 and LMP2), which are known for their poorer immunogenicity. Hence, these antigens exhibit weak propensity to induce antibody response and by NK cell killing through antibody-dependent cell cytotoxicity (ADCC).

EBV-mediated downregulation of HLA class I during lytic phase sensitizes infected cells to NK cells due to the absence of HLA class I (“missing self”) [27]. In B cells, the expression of HLA class I was recovered during EBV latency, which reduces the infected cells’ susceptibility to NK cells [27,64,65]. However, this was not observed in NPC (type II); diminished expression of HLA class I, like in lytic phase, was more common instead [28,29,81]. Whether this enhances susceptibility of NPC cells to NK is unclear.

Similarly, miR-BARTs can modulate host cell immune functions and contribute to immune evasion by dulling NK cell immune recognition through diminishing IL-12 release. These EBV-encoded miRNAs (miR-BARTs) are present at high abundance in NPC [82]. Mechanistically, miR-BARTs downregulate NK2GD ligands, MICA and MICB in NPC and diminish the cytotoxic potential of NK cells. This action was mediated by EBV-miR-BART7 and EBV-miR-BART2-5p [83,84,85]. From cytotoxicity assays, cells transduced with either EBV-miR-BART7 or EBV-miR-BART2-5p exhibited reduced sensitivity to NK killing compared to experimental controls. Secondly, miRNA-mediated downregulation of the cytokine IL-12 results in dampened NK cell cytotoxic response and IFN-γ production [77,79]. Site-directed mutagenesis results demonstrated that the *IL12B* reporter was directly targeted and repressed by at least five different EBV-miRNAs post infection [86,87]. In addition, EBV-miR-BART5 was demonstrated to target PUMA, a pro-apoptotic protein, hence, conferring resistance to NK-induced apoptosis [88]. EBV-miR-BART2 also inhibits lytic replication through targeted degradation of EBV DNA polymerase BALF5 [89].

**Table 1 viruses-13-00300-t001:** Summary of receptor-ligand interactions affecting NK-mediated restriction of EBV infection.

NK Receptors	Corresponding Ligand(s)	Key Findings	Source
**i. Killer Inhibitory Receptors**			
KIR2DL1–3, KIL2DL5, KIR3DL1–2, LILRB1	Classical HLA class I molecules: HLA-A, B, C	Downregulation of classical HLA class I molecules on the surface of infected cells is commonly observed during herpesvirus infection, including EBV.	[27,90,91]
		When the levels of classical HLA class I molecules are constant, changes in HLA-bound peptide (e.g., viral peptides on HLA-C) can abrogate the stimulation of NK inhibitory receptors (e.g., KIR2DL2/3).	[92,93,94]
NKG2A/CD94	HLA-E	(In latently infected B cells) HLA-E downregulation was observed together with downregulation of classical HLA class I molecules.	[27]
		CD56^bright^NKG2A^+^ NK cell subset was most potent in restricting outgrowth of EBV-infected B cells and LCLs, though it was unclear if NKG2A-HLA-E ligation was involved. There were, however, no significant differences in HLA-E expression between primary B cells and autologous LCL.	[76,79]
		Peptides derived from viral proteins may bind to HLA-E and reduce NK inhibition by affecting interaction with NKG2A/CD94.	[92,93,95]
**ii. Killer Activating Receptors**			
NKG2D	MICA/B, ULBP	During lytic phase of EBV infection, NK-mediated cytotoxicity of infected B cells was found to increase with expression of lytic cycle protein BZLF1, which induced upregulation of ULBP. ULBP is not expressed during latent infection.	[27,74]
		Higher frequencies of NKG2D^+^NKG2A^+^ NK cells were reportedly sensitive to autologous LCL, though it was uncertain if there were greater NKG2D stimulation.	[76]
		Low Mg^2+^ levels were demonstrated to decrease NK2GD expression in NK and T cells, impairing cytolytic responses against EBV. Restoration of Mg^2+^ levels increased NKG2D expression and reestablished EBV control.	[96]
NKG2C/CD94	HLA-E	NKG2C^hi^CD57^+^ NK subset was shown to persist during CMV but not EBV infection; NK lytic degranulation remained unchanged before and after EBV infection.	[97]
NKp30, NKp44, NKp80	B7-H6, BAT3 HLA-DP unknown	NK restriction of EBV-infected B cells was found to be partially mediated by NKp44 but not NKp30. Tonsillar NK cells were also found to express NKp44.	[79]
		NKp44 binds to HLA-DP, an HLA class II molecule. Coincidentally, HLA-DP is a co-receptor which EBV utilizes during viral entry into B cells.	[98,99]
CD16a (FcγRIIIa)	Fc of IgG (bound to target antigen on virus/infected cells)	A nucleotide substitution on the CD16a gene on NK cells was associated with recurrent viral infection, including EBV.	[100]
		The late lytic phase protein gp350/220, present on the surface of EBV-infected B cells, was found to increase susceptibility of infected cells to NK cell killing. ADCC via CD16a was proposed to be the possible mechanism; serum from EBV^+^ individuals were reportedly able to trigger NK cell degranulation and secretion of cytokines TNF-α and IFN-γ, while the EBV-negative serum did not. ADCC was also particularly potent against EBV-infected cells in the lytic phase.	[74,80]
		EBV-infected cells can release gp350^+^ particles that attach to bystander B cells and trigger antibody-dependent NK cell degranulation. IFN-γ production was markedly reduced, however, compared to TNF-α, and was accompanied by a significant reduction in NK-induced cell damage on target cells.	[74,80]
**iii. Co-stimulatory/inhibitory Receptors**			
4-1BB (CD137)	4-1BBL (CD137L)	Deficiency in 4-1BB, due to missense mutation, abolished receptor expression and ligand binding, leading to EBV-induced lymphoproliferation. This suggests the importance of 4-1BB in immune control of EBV infection in healthy individuals.	[101]
		4-1BBL is present on EBV-infected B lymphocytes. While 4-1BB deficiency resulted in poor cytotoxic T cell response in terms of proliferation, IFN-γ and perforin expression, it is unclear if NK cells were affected during 4-1BB deficiency.	[101]
DNAM-1 (CD226)	CD122 CD155	Enhanced NK killing of lytic EBV-infected cells was linked to upregulation of CD112 (with ULBP-1), as blocking of both ligands partially affected their susceptibility to NK.	[27]
		BZLF1 EBV lytic protein triggers DNAM-1-mediated NK activation possibly through an unidentified ligand, suggesting the importance of DNAM-1 in NK during NK infection.	[74]
OX40 (CD134)	OXOL (CD252)	Activated NK cells express OX40 and are stimulated to produce IFN-γ when cocultured with activated OX40^+^ pDCs.	[102]
		The roles of OX40 and OXOL are unknown in EBV, though OXOL expression have been reported on EBV-transformed B cell line.	
2B4 CD27	CD48 CD70	Primary immunodeficiencies, resulting from downstream mutation, led to problems in NK stimulation associated with EBV infection.	[69,70,71]
		Mutation in CD70 reportedly predisposes one to EBV-associated lymphoproliferation.	[70,71]

4-1BBL: 4-1BB ligand; ADCC: antibody-dependent cell cytotoxicity; B7-H6: B7 homolog 6; BAT3: HLA-B-associated transcript 3; BZLF1: BamHI Z fragment leftward open reading frame 1; CMV: cytomegalovirus; DNAM-1: DNAX accessory molecule 1; EBV: Epstein-Barr virus; gp350/220: glycoprotein 350/220; HLA A/B/C/E/DP: human leukocyte antigen A/B/C/E/DP; IFN-γ: interferon-γ; KIR2DL1/2DL5/3DL1-2/2DL2/2DL3: killer-cell immunoglobulin-like receptors 2DL1/2DL5/3DL1-2/2DL2/2DL3; LCL: lymphoblastoid cell lines; LILRB1: leukocyte immunoglobulin-like receptor subfamily B member 1; MICA/B: MHC class I polypeptide-related sequence A/B; NK: natural killer; NKG2A/2D/2C: natural killer group 2A/2D/2C; NKp30/44/80: natural killer cell p30/44/80-related protein; OX40L: OX40 ligand; pDCs: plasmacytoid dendritic cells; TNF-α: tumor necrosis factor-α; ULBP: UL16-binding protein.

**Table 2 viruses-13-00300-t002:** Summary of soluble mediators affecting NK-mediated restriction of EBV infection.

	Key Findings	Source
IFN-γ	NK cells present in the tonsils were found to produce more IFN-γ when cultured with activated DCs in comparison to NK cells from the blood. These tonsillar NK cells were able to reduce B cell transformation by EBV in vitro, highlighting the critical role of IFN-γ in NK cell cytotoxicity against EBV infection.	[79,103]
	The use of IFN-γ-blocking antibodies markedly reduced the inhibitory effect of tonsillar NK cells.	
	BZLF1 seems to inhibit the IFN-γ signaling pathway via downregulation of various downstream effects and IFN-γ receptor.	[104]
IL-12	IL-12 secretion was reported upon DC maturation with EBV viral stimuli. It subsequently caused NK cell proliferation and IFN-γ secretion to target EBV-mediated B cell transformation.	[77]
	A tonsillar anti-EBV NK subset exists, which readily secreted IFN-γ upon IL-12 stimulation in vitro.	[79]
TNF-α	A study suggested monitoring TNF-α levels as a useful prognostic marker, as TNF-α levels correlated with EBV infection status and EBV-associated peripheral T and NK cell lymphomas.	[105]
	In the presence of EBV-positive serum, NK cell degranulation and TNF-α production against EBV-infected cells were found to be triggered, demonstrating enhanced activation of NK cells during the EBV lytic phase.	[80]
IL-10	Promote NK cell proliferation and activation in vitro and enhance antiviral innate immunity through inhibition of activation-induced death in NK cells.	[106,107]
	IL-10 serum levels were significantly higher during chronic EBV disease relative to controls, suggesting that IL-10 could play a role in EBV progression. In EBV infection mouse models, viral IL-10 seemed to contribute to acute infection by inhibiting NK cell-mediated killing of infected B cells.	[75]
	IL-10 could induce LMP1 expression in tonsillar B cells infected with an EBNA-2-deficient EBV strain and enhance LMP1 expression in EBV-positive NK lymphoma cell lines, indicating that IL-10 possibly contributes to type II EBV latency.	[108]
	Knockdown of IL-10 induced EBV lytic infection and replication in EBV-associated tumors, further highlighting the role of IL-10 in maintaining long-term EBV latent infection.	[109]
IL-18	IL-18 levels significantly increase during EBV infection. In a study conducted by our team, we found elevated levels of IL-18 in NPC tissues compared to control tissues. These elevated IL-18 levels may enhance functional exhaustion of NK cells in NPC through upregulation of PD-1.	[110]
Perforin	Mutations that impair maturation of perforin is associated with chronic active EBV infection, providing evidence of cell-mediated cytotoxicity in controlling EBV infections	[111]

BZLF1: BamHI Z fragment leftward open reading frame 1; DCs: dendritic cells; EBNA-2: Epstein-Barr nuclear antigen 2; EBV: Epstein-Barr virus; IFN-γ: interferon-γ; LMP1: latent membrane protein 1; NK: natural killer; NPC: nasopharyngeal carcinoma; PD-1: programmed cell death protein 1; TME: tumor microenvironment; TNF-α: tumor necrosis factor-α.

EBV proteins expressed during type II latency (eg LMP1, LMP2A/B, vBcl2 and vIL10) can contribute to NK evasion. Latent protein LMP1 can contribute to NK evasion in several ways. First, LMP1 allows EBV-infected cells to survive despite being co-cultured with p TGF-β1, which is a known NK-inhibiting cytokine [112,113,114]. LMP1 does so through the upregulation of inhibitor of differentiation-1 (Id-1), which negates effects of TGF-β1 against NPC [115]. Second, LMP1 may allow infected cells to resist NK-induced cell death since they induce expression of several anti-apoptotic proteins such as Survivin, A20, Bcl2 [116] and prosurvival receptor, 4-1BB [117,118] in EBV-infected cells. Upregulation of anti-apoptotic proteins will enable infected cells to resist apoptosis induced by NK cells via death receptors e.g., Fas and TRAIL, which is proposed to be the main mechanism of how NK targets NPC cells [119].

EBV-encoded vIL10 prevents lytic reactivation in EBV-infected cells, thus reducing their susceptibility to NK cells. vIL10 was also found to stabilize LMP1 expression in EBV-infected tonsillar B cells, thus promoting latency [108]. In addition, deactivation of vIL-10 was found to reactivate lytic phase which sensitizes infected cells to NK elimination [75,109]. Finally, vBcl2, an EBV-encoded anti-apoptotic protein, can resist NK-induced apoptosis [74]. LMP2A and LMP2B can limit anti-viral response against EBV-infected cells through downregulation of IFN signaling [120], and hence, suppress cytokine-mediated anti-viral effects. LMP2A also induces the downregulation of NKG2D and MICA/B ligands on NK cells. This is mediated by protein disulfide isomerase, which disrupts disulfide bonds, and hence, modifies protein conformation and results in reduced surface expression of MICA/B [29].

## 6. What Is Currently Known about NK Cells in NPC?

Analysis of peripheral blood lymphocytes between NPC patients and healthy individuals revealed no differences in NK cell abundance or percentage based on analysis of CD56^+^CD3^-^ in two studies [121,122]. Another study, however, noted a higher percentage of circulating NK cells in NPC patients compared to healthy controls [112]. With regards to NK cell phenotypes, one study noted no difference in terms of percentage of NK cells positive for NKG2D (NK activating) and KIR2DL2/DL3 (CD158b) and NK2GA (CD159a) (both NK inhibitory) between NPC and healthy controls [122]. However, a lower percentage of NK cells expressing NKp30 and NKp46 (activating receptors) was identified in NPC patients than in healthy controls [122]. In another study, a lower percentage of activated NK cells (NKG2D and Granzyme B positive) was noted in NPC patients, while a higher percentage of NK cells expressing inhibitory ligands (KIR2DL2/KIR3DL3) was seen in NPC patients than healthy controls. These findings indicated poorer functional NK cells in NPC patients than in healthy controls [112]. The list of relevant NK receptors and ligands reported in NPC is summarized in Table 3.

Complementary and coaptation of peripheral NK cells with adaptive EBV-specific CTL and γδ-CTL can occur in NPC patients [121,123,124]. Zheng and colleagues noted that the stimulation of PBMC from NPC patients induced more robust NK expansion than healthy controls or NPC patients who are in complete remission of more than five years [121]. Intriguingly, patient-derived expanded NK cells induced greater cytotoxicity against NPC cells than those from healthy controls and long-term survivors [121]. In contrast, γδ-CTLs expanded from NPC patients exhibited poorer cytotoxicity against the same NPC cells [121] Furthermore, when supernatants of NK cell cultures were assessed for cytokine(s) production, those of cultured NK cell from NPC patients had the lowest production of IFNγ but higher production of IL-12, IL-15 and TNF-α. Higher production of cytokines that promote NK cell growth ex vivo may have explained the greater functionality of cultured NK from patients.

In another study, Zhao and colleagues showed that diminished NKG2D-positive NK population could be induced by overexpression of metastasis-associated colon cancer-1 (MACC1) on NPC [112]. They showed that the overexpression of MACC1 on NPC cells promoted increased expression of TGF-β1, a potent inhibitor of NK cells, which downregulates NKG2D [112]. However, the experiments were performed on HONE1 cell line, which was reportedly an NPC cell line contaminated with HeLa cells [125].

It is important to note that NK cells do not interact solely with NPC cells. NK cells in NPC have been reported to interact with macrophages [126,127], dendritic cells [127] and dysfunctional CD8 T cells [127]. NK–tumor interactions included HLA-C/KIR2DL4, HLA-E/CD94 (KLD1), TIM3 (HAVCR2)/Galectin-9, CD96-PVR/Nectin1 and TIGIT-PVR/Nectin2 [126,127]. HLA-E/CD94 interaction is of notable interest, since HLA-E is a ligand of NKG2A, which is a receptor expressed by NK cells that is involved in EBV restriction [65,76,79]. Additionally, TIM3-Galectin-9 and TIGIT-Nectin2 interactions were also noted between NK cells and dysfunctional CD8 cells, macrophages and dendritic cells, respectively [126,127].

In summary, so far, studies of the NK cells in NPC are less conclusive compared to those of NK cells in EBV-infected B cells (Figure 1). While increased susceptibility to NK cells was noted for infected B cells during lytic phase (Figure 1a), the susceptibility to NK for infected epithelial cells during lytic phase and lytic reactivated NPC cells to NK cells remains unclear (Figure 1c). Additionally, it is also unclear if the same peripheral and tonsillar NK subsets can restrict EBV-infected epithelial cells similarly (Figure 1c,d), as noted in EBV-infected B cells (Figure 1a,b). Strikingly, differences in HLA class I expression (upregulation in latently infected B cells versus downregulation in NPC) suggest possible inherent differences in NK evasion mechanisms which could be attributed by tumor-based mechanisms instead of EBV itself. Further functional and phenotype characterization of NK subsets unique to EBV restriction in B cells could be carried out in an NPC context to address how changes in NK functionality or susceptibility to NK elimination contribute to NPC tumorigenesis.

## 7. Translational Applications of NK Cells in NPC

A deeper understanding of NK cell biology in NPC can pave translational clinical advancement in NPC management. Our lab previously reported that NK cells from NPC patients can induce ADCC of C666-1 NPC cells in the presence of cetuximab, a monoclonal antibody against epidermal growth factor receptor (EGFR) [128,129]. Furthermore, ADCC of NPC cells can be enhanced after an overnight treatment of primary NK cells from healthy control or NPC patients with Poly-ICLC, a Toll-like receptor 3 (TLR3) agonist [128]. Poly-ICLC-treated NK cells, regardless of CD16 (FcɣRIIIa) genotype, exhibited greater ADCC against C666-1 NPC cells. Additionally, Poly-ICLC-treated NK cells underwent more significant degranulation and IFNγ secretion, and showed a greater percentage of NK cells expressing CD107a, granzyme B and IFNγ, after co-culture with C666-1 and cetuximab. Separately, patients’ NK cells can be activated and expanded using irradiated K562-mb15-41BBL, which triggered robust expansion of CD16^+^ NK cells [129]. Concurrent treatment of these expanded autologous NK cells with cetuximab in patients with refractory/metastatic NPC were well tolerated with no grade 3–4 toxicities in this phase I clinical trial.

Enhancement of functional NK cells has been shown to be augmented by the use of Interferon beta (IFNβ), checkpoint blockade of PD-1/PD-L1 and chemotherapies such as cisplatin, 5-fluorouracil and gemcitabine [119,130,131]. Makowska and colleagues reported that IFNβ treatment of NK cells induced tumor necrosis factor apoptosis inducing ligand (TRAIL) expression and enhanced their cytotoxicity against NPC cells [130]. Furthermore, IFNβ triggers soluble TRAIL (sTRAIL) secretion; serum levels of sTRAIL increased in NPC patients under IFNβ maintenance therapy [130]. Concentrated sTRAIL from patient’s serum could also trigger apoptosis of NPC from patient-derived xenograft ex vivo [130]. Although IFNβ could induce expression of PD-1 and PD-L1 on NK cells and NPC cells, respectively, it can be overcome with antibody-mediated blockade of PD-1/PD-L1 [119]. Finally, PD-1/PD-1 blockade can also sensitize NPC cells to NK cells after treatment with cisplatin, 5-fluorouracil and gemcitabine [131].

**Table 3 viruses-13-00300-t003:** Summary of receptor-ligand associated NK immune evasion mechanisms by NPC and other cancers.

NK Receptors	Corresponding Ligand(s)	Key Findings	Source
**i. Killer Inhibitory Receptors**			
KIR2DL4 KIR2DL2/DL3	HLA-C	HLA-C and KIR2DL4 was one of the more significant immune cell-tumor interactions (on NK) during single-cell transcriptomic analysis of NPC.	[126]
		No difference in expression of KIR2DL2/DL3 (CD158b) on peripheral NK cells between NPC patients and healthy control.	[122]
NKG2A/CD94	HLA-E	HLA-E and CD94 (KLRD1) was another one of the more significant immune cell-tumor interactions (on NK) during single-cell transcriptomic analysis of NPC.	[126]
		Upregulation of peptide-loaded HLA-E, the ligand of CD94-NKG2A inhibitory receptor complex, was observed in tumor cells following exposure to IFN-γ.	[132,133]
		Expression of *KLRC1* gene, which encodes NKG2A, was found to correlate with HLA-E in tumor samples.	[134]
		Enrichment of NKG2A-expressing NK cells was also observed in the TME.	[135]
		Overexpression of HLA-E is observed in several cancers and linked to poor outcome.	[135,136,137]
		No difference in expression of NKG2A (CD159a) on peripheral NK cells between NPC patients and healthy control.	[122]
LILRB1 KIR2DL4	HLA-G	High expression of HLA-G predicted poor survival, treatment failure and distant metastases in NPC. HLA-G was detected on 79.2% of 522 NPC specimens but not normal nasopharyngeal tissue.	[138]
		Plasma levels of sHLA-G are significantly increased in patients of many malignancies.	[139,140]
		HLA-G was found to inhibit proliferation and cytotoxicity of peripheral blood NK cells, as well as inhibiting NK cell chemotaxis through downregulation of chemokine receptors expression.	[141,142,143,144]
		HLA-G expression is also linked with poorer prognosis in breast and ovarian cancers.	[136,137]
LILRB1 LILRB2 KIR3DL1/2	HLA-F	High HLA-F expression was associated with local recurrence-free survival and distant metastasis-free survival. NPC patients also had higher soluble HLA-F in plasma than normal controls.	[145]
**ii. Killer Activating Receptors**			
NKG2D	MICA/B ULBP	EBV-encoded microRNAs, EBV-miR-BART7 and EBV-miR-BART2-5p, were found to promote the downregulation of these MICA/B; hence, reducing detection by NK cells.	[83,84]
		Downregulation of MICA/B, mediated by protein disulfide isomerase, was reportedly observed in LMP2A-expressing epithelial cells, though there was also HLA-ABC downregulation through promoter hypermethylation.	[29]
		NKG2D was downregulated on NK through overexpression of MACC1 on NPC, which enhanced secretion of TGF-β1—a potent inhibitor of NK cells.	[112]
		Expression of IDO, an enzyme involved in tryptophan degradation pathway, was reportedly observed on both tumor cells and macrophages in tumor stroma of NPC. IDO metabolites, also secreted by EBV-infected cells, could inhibit NKG2D expression, and consequently, NKG2D-mediated cytotoxicity of NK cells.	[146,147]
		MICA and MICB may be shed by tumors through exosomes or by metalloproteinase cleavage.	[148,149,150,151,152]
		Prolonged NKG2D signaling can desensitize NK activating ligand stimulation through downregulation of DAP10 and DAP12.	[153]
		NK cells may be desensitized by endothelial cells or myeloid cells expressing NK2GD ligands in the stroma.	[154,155]
NKp30, NKp46	B7-H6, BAT3 unknown	Lower percentages of NKp30^+^ and NKp46^+^ NK cells were observed respectively in peripheral NK cells from NPC patients than that of healthy controls.	[122]
**iii. Co-inhibitory Receptors**			
PD-1 CTLA-4 LAG3 HAVCR2 (TIM3)	PD-L1 CD80/86 HLA class II Galectin-9	High PD-1 expression, especially with co-expression of PD-L1, was associated with high local recurrence and unfavorable clinical outcome in stage IV M0 NPC.	[156]
		PD-1 was found expressed on NK cells present in the tumor-infiltrating immune cells in NPC; PD-1 expression on NK cells was notably induced by IL-18, which was significantly higher in NPC biopsy than in normal nasopharynx.	[110]
		PD-L1 in soluble form can be detected in the plasma of NPC patients; soluble PD-L1 levels in plasma were found to positively correlate with clinical stage and N stage.	[157]
		PD-1, CTLA-4, LAG3 and TIM3 are also expressed in some NK cells, which may dampen their response in the presence of corresponding ligands.	[20,158,159]
		Blockade of PD-1/PD-L1 interaction may circumvent inhibition of NK antitumor activity, suggesting the role of PD-1 ligation in blocking NK cell functionality in TME.	[160]
		HAVCR2 (TIM3) of NK interacts with Galectin-9 on tumor cells, dysfunctional CD8 cells, macrophages and dendritic cells in NPC.	[126,127]
CD96 TIGIT	PVR/Nectin1 PVR/Nectin2	CD96-PVR/Nectin1 and TIGIT-PVR/Nectin2 interactions between NK and tumor cells were detected in NPC. TIGIT-Nectin2 interactions were also detected between NK and macrophages/dendritic cells in NPC.	[126,127]

CTLA-4: cytotoxic T-lymphocyte associated protein 4; DAP10/12: DNAX-activation protein 10/12; EBV-miR-BART7/2-5p: EBV-encoded microRNA BART7/2-5p; HAVCR2: hepatitis A virus cellular receptor 2 homolog; HLA A/B/C/E/G/F: human leukocyte antigen A/B/C/E/G/F; IDO: indoleamine 2,3-dioxygenase; IFN-γ: interferon-γ; KIR2DL4/3DL1/3DL2: killer-cell immunoglobulin-like receptors 2DL4/3DL1/3DL2; KLRC1: killer cell lectin like receptor C1; KLRD1: killer cell lectin-like receptor subfamily D, member 1; LAG3: lymphocyte-activation gene 3; LILRB1/2: leukocyte immunoglobulin-like receptor subfamily B member 1/2; LMP2A: latent membrane protein 2; MACC1: metastasis-associated colon cancer-1; MICA/B: MHC class I polypeptide-related sequence A/B; NK: natural killer; NKG2A/2D: natural killer group 2A/2D; NPC: nasopharyngeal carcinoma; PD-1: programmed cell death protein 1; PD-L1: programmed death-ligand 1; PVR: Poliovirus Receptor; sHLA-G: soluble HLA-G; TGF-β1: transforming growth factor β1; TIGIT: T cell immunoreceptor with Ig and ITIM domains; TIM3: T-cell immunoglobulin and mucin domain 3; TME: tumor microenvironment; ULBP: UL16-binding protein.

The complex interactions between NK cells and NPC tumor microenvironment can also help prognosticate NPC. Observation from single-cell RNA sequencing (scRNA seq) analysis of NPC showed positive correlation of NK immune signatures with improved progression-free survival in NPC patients [127]. This NK signatures comprised of a gene expression panel of 13 genes such as genes expressing perforin (*PRF1*), granulysin (*GNLY*), NKG2A (*KLRC1*), NKG2C (*KLRC2*) and CD94 (*KLRD1*) [127]. In contrast, another study reported poorer two-year survival in NPC patients with a higher percentage of CD56^+^ cells, as compared to those with lower CD56^+^ percentage on immunohistochemistry (IHC) [110]. As this study had examined CD56 as a single parameter due to the limitation of IHC, the use of multi-fluorescence IHC can overcome the difficulty of distinguishing between NK cell versus NKT cell population as well as allowing the evaluation of other surface marker co-expression on NK cells (e.g., CD56 and CD16), so as to better characterize different NK subsets. Finally, these observations should be further validated through paired multiparameter flow analysis to ascertain the actual phenotype of NK in terms of NK subset (CD56^bright^CD16^dim^ vs. CD56^dim^CD16^+^) and expression of other NK receptors (e.g., NKG2A) reported in the earlier studies [74,76,79].

## 8. Future Directions

Our knowledge of viral–host mechanisms between EBV and NK-mediated anti-viral immunity are largely derived from experiments utilizing B lymphocytes, which are immortalized after EBV infection. Models of EBV-infected epithelial cells are difficult to establish, since EBV infection does not immortalize these epithelial cells in vitro. EBV-infected epithelial model had to be derived from immortalized/modified epithelial cells [42,44]. Until recently, orthotropic epithelial model presents a more accurate representation of stratified epithelium, which allows replicative lytic EBV infection [38,43,161]. Susceptibility of these infected epithelial cells to NK cells should be reexamined to reaffirm the interactions observed from earlier B cell experiment models. EBV-challenged orthotropic epithelial model could lay the groundwork for modeling NPC tumorigenesis in the presence of additional factors such as immune cells and inflammatory factors.

NPC tumorigenesis is a complex process which is not induced by EBV alone. Other parameters including EBV subtypes, host genetic factors and environmental triggers have been positively associated with the risk of NPC in endemic regions. Changes which stabilize EBV latency, promote immune evasion, reduce lytic reactivation and hamper the host’s immune fitness should all be considered when one investigates the role of NK cells in EBV-associated NPC. In addition, viral–host interactions, which determine the susceptibility of EBV-infection to NK-mediated surveillance, should be further addressed if they influence risk of developing NPC.

Finally, NK cells in the peripheral blood and tissues are different in terms of abundance, maturity and functionality. Given that EBV and NPC affect the tonsils, the contribution of tissue-resident NK cells may play a larger role in both disease contexts compared to that of peripheral NK cells. Importantly, the mode of EBV restriction could be different because of the difference in abundance of NK. Incidentally, the tonsils also contain intraepithelial innate lymphoid cells (ILC) that are very similar to NK cells. An example is the ILC1, which also produces high levels of IFN-γ, similar to NK cells, though largely independent of IL-15Rα, unlike NK cells [162]. This reported phenotype of ILC is very similar to the tonsillar NK cells as marked by expression of NKp44, NKp46, NK1.1 and CD160 [79,162]. Therefore, a clear distinction between NK and ILCs are required during the analysis of tissue-resident NK in the respective disease pathogenesis. More emphasis should be placed on the analysis of tonsillar NK cells and their functionality during EBV infection and NPC, then determine how changes in their NK phenotype may influence NPC risk.

## Figures and Tables

**Figure 1 viruses-13-00300-f001:**
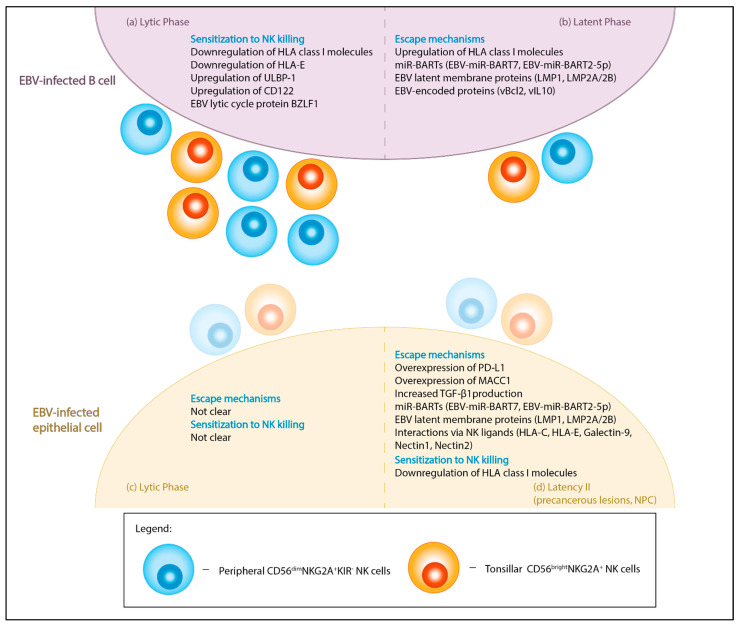
Graphical summary of the interactions between NK cells and EBV-infected cells. EBV infects both B cells (**a**,**b**) and epithelial cells (**c**,**d**) in either lytic or latent forms. Peripheral CD56^dim^NKG2A^+^KIR^-^ NK cells (blue) and tonsillar CD56^bright^NKG2A^+^ NK cells (orange) have been identified to be the main NK subset in restricting EBV-infection in B lymphocytes. The reactivity of these subsets against EBV-infected epithelial cells remains unclear. (**a**) During the lytic phase, EBV-infected B lymphocytes are more sensitized to NK killing due to changes in the expression of NK receptor ligands (HLA class I molecules, HLA-E, ULBP-1, CD122) and induction of lytic cycle protein BZLF1. (**b**) During the latent phase, there is greater NK evasion in infected B cells due to the expression of miR-BARTs, latent membrane antigens (LMP1, LMP2A/B) and other virus-encoded proteins (vBcl-2, vIL10). The miR-BARTs and LMP2A downregulate NKG2D ligands and diminish NK activation potential. The anti-apoptotic protein, vBcl2, provides protection against NK-mediated apoptosis, while pre-latent phase protein vIL-10 prevents lytic reactivation in EBV-infected tonsillar B cells. Additionally, the upregulation of HLA class I molecules can dampen NK activation through the engagement of inhibitory receptors. (**c**) EBV lytic infection in normal epithelial cells leads to cytopathy, while lytic reactivation in NPC results in genomic instability. However, how this affects evasion or sensitization to NK cells remains unclear. (**d**) In type II latency, EBV-infected epithelial cells can escape NK attack through the overexpression of PDL-1, which promote NK exhaustion and/or the overexpression of MACC1 which leads to increased TGF-β1 production. Additionally, expression of miR-BARTs, latent membrane proteins and NK inhibitory ligands (HLA-C, HLA-E), and ligation of co-inhibitory receptors on NK (Galectin-9-TIM3, Nectin1-CD96 and Nectin2-TIGIT) may also desensitize antitumor NK responses. In contrast, the downregulation of HLA class I molecules, common in NPC, may sensitize tumor cells to NK cells due to lack of NK inhibitory ligands (“missing-self”). HLA, human leukocyte antigen; ULBP-1, UL16 binding protein 1; BZLF1, BamHI Z fragment leftward open reading frame 1; BART, BamHI-A rightward transcripts; LMP, latent membrane protein; MACC1, metastasis-associated colon cancer-1; vBcl2, viral Bcl-2 homolog; vIL10, viral interleukin-10; PDL-1, programmed cell death protein ligand 1, TGF-β1, transforming growth factor-β 1; NKG2, natural killer group 2; TIM3, T-cell immunoglobulin and mucin domain 3; TIGIT, T cell immunoreceptor with immunoglobulin and ITIM domains.

## Data Availability

No new data were created or analyzed in this study. Data sharing is not applicable to this article.

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
