# Peer review of "The Role of NK Cells in EBV Infection and EBV-Associated NPC"

_viruses, 2021, doi:10.3390/v13020300_

Round 1

Reviewer 1 Report

The authors review the role of Natural Killer cells in controlling infection by Epstein-Barr virus and nasopharyngeal carcinoma. Given the importance of NK cells in EBV infection, the present review poses a timely opportunity to synthesize information regarding NK cell function during NPC. The review is well written, and most parts, especially the introductory sections, are clear and focussed. However, the review loses focus towards the end, and readers may benefit from a stronger effort by the authors to connect NK cell-NPC findings into broader biological trends, perhaps with a figure.

Major criticisms

Sections 1-5 provide the appropriate background to understand the role of NK cells during EBV infection and NPC. However, after that point the review loses focus on the NK cell-NPC relationship. While Section 6 is titled “6. How do non-viral factors contribute to NK evasion in NPC?”, the literature discussed is only tangentially related to NK cells, with very few concrete links between genetic variants associated with NPC and NK cell function – this is especially true for the last two paragraphs of section 6. Section 7 is better focussed on the review topic but alternates between discussing the biology of NK cell-NPC interactions (paragraphs 1-3, 6) and possible translational approaches for immunotherapies/biomarkers (paragraphs 4-5, 7) – each paragraph seems focussed on a different observation, without synthesis by the authors to connect each paragraph. I think two changes to section 6 and 7 may be of use: 1) Section 6 could be removed from the manuscript, and the current section 7 split into two distinct sections: the involvement of NK cells during NPC and how NK cell biology can be harnessed for translational/therapeutic approaches. 2) The paragraphs outlining NK cell responses to NPC in the current section 7 (paragraphs 1-3, 6) could be restructured in a way that better connects these sometimes-disparate observations. I understand that this may not be completely possible, but even a sentence at the end of one paragraph that leads to the next may help readers make connections.

I do not think the figure in the paper reflects the thesis of the review, and I don’t think it is very useful – i.e. I don’t think it is showing anything that the readers would have had trouble digesting by just reading the paper. I would recommend changing Figure 1 to a graphical summary of the most important aspects of the NK cell interface with NPC. For example, perhaps a summary of how NK cells interact with healthy tissues, with EBV-infected B cells/oropharynx, with premalignant lesions, and with NPC of various stages.

Minor criticisms

  • Where are references 3 and 4? We skip from ref 2 (line 33) to ref 5 (line 39)
  • Lines 74 – 76: I thought the oropharynx and nasopharynx were two separate parts of the throat. The authors seem to imply there are nasopharyngeal cells in the oropharynx? Is this true?
  • In section 2, perhaps worth dedicating a section to the role of lytic reactivation in promoting NPC – for example, EBV copies in the blood can be increased with disease onset, EBV lytic reactivation leads to VEGF production, and can lead to genome instability.
  • In section 3, the naming convention of KIR and KAR should be changed, as this is confusing – KIR is usually used to refer to a specific set of inhibitory and activating receptors Killer cell Immunoglobulin-like R Using it as an acronym for all Killer Inhibitory Receptors, and including non-KIR receptors in this group, will unnecessarily complicate things.
  • Line 152: Although the authors state Pappworth et al as the first to show NK cell recognition of the lytic cycle, there was already strong evidence in place from Blazar et al 1980, JEM (PMID 6244358).
  • Line 178: The authors state that there is only one study, to their knowledge, on tonsillar NKs role in EBV infection. There are more: e.g. Strowing et al 2008 Plos Pathog (PMID: 18266470) and Lünemann et al 2013 J Immunol (PMID: 24108698), and there may well be more. These studies showed a slightly different phenotype in immunoresponsive tonsillar NKs (e.g. they are KIR+). Discussion that unites these three studies would be useful, and the authors should do a more extensive literature search given the likely importance of tonsillar NK cells in regulating EBV infection.
  • The title of section 5: “How does type II EBV latency contribute to NK cell evasion?” may raise questions about what the different types of latency are. A brief description of EBV latency programs, and how latency II relates specifically to NPC would help readers understand this naming convention.
  • In section 5, the authors should discuss EBV-mediated downregulation of HLA, how this changes through lytic and latent stages, and that HLA class I is expressed during latency (or is it expressed during latency II?), which helps shield these infected latent cells from NK cell lysis.
  • Lines 281-310 are not related to NK cells. The preceding paragraphs of this section are also only vaguely connected to NK cells. If the authors want to discuss the impact of genetic variation on NK cells and NPC, there are several studies (e.g. by Mary Carrington’s group) that have found significant associations between NPC or EBV and KIR, or KIR ligands in HLA, that would be more relevant to discuss.

Author Response

Address of comments from reviewer #1

We thank reviewer #1 for his/her kind comments and valuable suggestions to improve our manuscript. We have gone on to address all the comments posted by reviewer #1 as highlighted below.

  1. Sections 1-5 provide the appropriate background to understand the role of NK cells during EBV infection and NPC. However, after that point the review loses focus on the NK cell-NPC relationship. While Section 6 is titled “6. How do non-viral factors contribute to NK evasion in NPC?”, the literature discussed is only tangentially related to NK cells, with very few concrete links between genetic variants associated with NPC and NK cell function – this is especially true for the last two paragraphs of section 6. Section 7 is better focussed on the review topic but alternates between discussing the biology of NK cell-NPC interactions (paragraphs 1-3, 6) and possible translational approaches for immunotherapies/biomarkers (paragraphs 4-5, 7) – each paragraph seems focussed on a different observation, without synthesis by the authors to connect each paragraph. I think two changes to section 6 and 7 may be of use: 1) Section 6 could be removed from the manuscript, and the current section 7 split into two distinct sections: the involvement of NK cells during NPC and how NK cell biology can be harnessed for translational/therapeutic approaches. 2) The paragraphs outlining NK cell responses to NPC in the current section 7 (paragraphs 1-3, 6) could be restructured in a way that better connects these sometimes-disparate observations. I understand that this may not be completely possible, but even a sentence at the end of one paragraph that leads to the next may help readers make connections.

>> We thank the reviewer for the suggestions.

With regards to section 6, we have considered the reviewers’ suggestion and removed this section as suggested.

With regards to section 7, we have segregate both sections into the 2 corresponding sections: Section 6 is now “What is currently known NK cells during NPC?” and Section 7 is now “Translational applications of NK cells in NPC”.  

  1. I do not think the figure in the paper reflects the thesis of the review, and I don’t think it is very useful – i.e. I don’t think it is showing anything that the readers would have had trouble digesting by just reading the paper. I would recommend changing Figure 1 to a graphical summary of the most important aspects of the NK cell interface with NPC. For example, perhaps a summary of how NK cells interact with healthy tissues, with EBV-infected B cells/oropharynx, with premalignant lesions, and with NPC of various stages.

>> We thank the reviewer for the suggestions. We agree that a graphical summary will be more appropriate for better clarity to the readers and have hence replaced figure 1 which summarises the key NK findings with EBV-infected B cells and epithelial cells in lytic and latent phase.

  1. Where are references 3 and 4? We skip from ref 2 (line 33) to ref 5 (line 39)

>> We apologise for missing references. We have added references 3 and 4 which was accidentally omitted at line 33-34.

  1. Lines 74 – 76: I thought the oropharynx and nasopharynx were two separate parts of the throat. The authors seem to imply there are nasopharyngeal cells in the oropharynx? Is this true?

>> We apologise for the confusion. As pointed by the reviewer, the oropharynx and nasopharynx are 2 different parts of the throat. For better clarity, we modified lines 72-74 to “…both B lymphocytes and epithelial cells. Infection of both cell types in the pharynx such as the tonsils, have been shown to be crucial for viral persistence in the host30,31

  1. In section 2, perhaps worth dedicating a section to the role of lytic reactivation in promoting NPC – for example, EBV copies in the blood can be increased with disease onset, EBV lytic reactivation leads to VEGF production, and can lead to genome instability.

>> We thank the reviewer for the suggestion. We have included a new paragraph on the role of lytic reactivation in NPC as suggested.

  1. In section 3, the naming convention of KIR and KAR should be changed, as this is confusing – KIR is usually used to refer to a specific set of inhibitory and activating receptors Killer cell Immunoglobulin-like RUsing it as an acronym for all Killer Inhibitory Receptors, and including non-KIR receptors in this group, will unnecessarily complicate things.

>> We thank the reviewer for the suggestion. We agree that the naming of KIR and KAR can be confusing. We have replaced KAR and KIR abbreviations with “activating receptors” and “inhibitory receptors” respectively.

  1. Line 152: Although the authors state Pappworth et al as the first to show NK cell recognition of the lytic cycle, there was already strong evidence in place from Blazar et al 1980, JEM (PMID 6244358).

>> We thank the reviewer for pointing out the error. We have corrected the sentence and included PMID 6244358 in the paragraph.

  1. Line 178: The authors state that there is only one study, to their knowledge, on tonsillar NKs role in EBV infection. There are more: e.g. Strowing et al 2008 Plos Pathog (PMID: 18266470) and Lünemann et al 2013 J Immunol (PMID: 24108698), and there may well be more. These studies showed a slightly different phenotype in immunoresponsive tonsillar NKs (e.g. they are KIR+). Discussion that unites these three studies would be useful, and the authors should do a more extensive literature search given the likely importance of tonsillar NK cells in regulating EBV infection.

>> We thank the reviewer for pointing out the discrepancy. We acknowledge the missing information and have included the summary of the quoted sources in the paragraph on tonsillar NK cells and their role in EBV infection.

  1. The title of section 5: “How does type II EBV latency contribute to NK cell evasion?” may raise questions about what the different types of latency are. A brief description of EBV latency programs, and how latency II relates specifically to NPC would help readers understand this naming convention.

>> We thank the reviewer for the suggestion. We have included an introductory paragraph in section 5 as recommended for better clarity.

  1. In section 5, the authors should discuss EBV-mediated downregulation of HLA, how this changes through lytic and latent stages, and that HLA class I is expressed during latency (or is it expressed during latency II?), which helps shield these infected latent cells from NK cell lysis.

>> We thank the reviewer for the suggestion. We have included the discussion of EBV-mediated changes to HLA during lytic and latent stages in a new paragraph. As pointed out, HLA class I was noted to be downregulated during type II latency (in NPC) unlike latently infected B cells.

  1. Lines 281-310 are not related to NK cells. The preceding paragraphs of this section are also only vaguely connected to NK cells. If the authors want to discuss the impact of genetic variation on NK cells and NPC, there are several studies (e.g. by Mary Carrington’s group) that have found significant associations between NPC or EBV and KIR, or KIR ligands in HLA, that would be more relevant to discuss.

>> We thank the reviewer for the suggestion. We have excluded section 6 based on your earlier suggestion and replaced section 6 with part I of section 7.

Reviewer 2 Report

viruses-1093307-peer-review-v1viruses-1093307-peer-review-v1

This is a well written review on the role of NK cells in EBV associated cancers with the focus on NPC. There are several issues that could enhance the readability and impact.

Line 11: change patients to “infected individuals”

Line 34-35: rephrase the sentence as it seems missing a few words. “acquired late in” -> late in life

Line 35-36: provide citations (e.g. PMID: 31481499, PMID: 31448229)

Line 63: additional more recent citations (e.g. PMID: 31481499, PMID: 31295651, PMID: 32737537)

Line 106: the more accurate number can be cited from (PMID: 26013798)

Line 128: change “high expression of HLA Class I expression in ” -> “high expression of HLA Class I in”

Line 131: please clarify what “this subset” refers to in the sentece.

Section 4 nicely summarizes the role of NK in restricting EBV infection mainly focusing on B cell restriction. It also discusses the NK dysfunction in specific diseases and its association with EBV tumors. It would be important to also discuss whether NK cells in NPC are “dysfunctional”, and also discuss whether there are literature on NK cells restricting EBV+ epithelial cells.

Section 5: It would be more logical to add a short paragraph at the beginning of this section reminding readers that Type II latency program is associated with lower BZL1 expression, higher miR-BARTs, and higher LMPs and each of these could have a role in NK evasion.

It would be also interesting to discuss therapeutics to enhance NK cells and their potential benefit to NPCs (e.g PMID: 26000725)

Author Response

Address of comments from reviewer #2

We thank reviewer #2 for his/her kind comments and valuable suggestions to improve our manuscript. We have gone on to address all the comments posted by reviewer #2 as listed below.

  1. Line 11: change patients to “infected individuals”

>> We thank the reviewer for the suggestion. We have replaced “patients” with “infected individuals” at line 11.

  1. Line 34-35: rephrase the sentence as it seems missing a few words. “acquired late in” -> late in life

>> We apologise for the error as pointed out by the reviewer. We have corrected the sentence to “acquired late in childhood” and included the missing references.

  1. Line 35-36: provide citations (e.g. PMID: 31481499, PMID: 31448229)

>> We thank the reviewer for the suggestion. We have included both recommended citations in line 35-36 as suggested.

  1. Line 63: additional more recent citations (e.g. PMID: 31481499, PMID: 31295651, PMID: 32737537)

>> We thank the reviewer for the suggestion. We have included the more recent citations at line 63 as suggested.

  1. Line 106: the more accurate number can be cited from (PMID: 26013798)

>> We thank the reviewer for the suggestion. We have included the citation in line 106 as suggested.

  1. Line 128: change “high expression of HLA Class I expression in ” -> “high expression of HLA Class I in”

>> We thank the reviewer for the suggestion. We have modified the sentence to “high expression of HLA Class I in” at line 128 as suggested.

  1. Line 131: please clarify what “this subset” refers to in the sentece.

>> We apologise for the confusion. This subset refers to CD56dimNKG2A+KIR- NK cells. We have hence modified “lower frequencies of this subset” to “lower frequencies of CD56dimNKG2A+KIR- NK cells” at line 131 for better clarity.

  1. Section 4 nicely summarizes the role of NK in restricting EBV infection mainly focusing on B cell restriction. It also discusses the NK dysfunction in specific diseases and its association with EBV tumors. It would be important to also discuss whether NK cells in NPC are “dysfunctional”, and also discuss whether there are literature on NK cells restricting EBV+ epithelial cells.

>> We thank the reviewer for the suggestion. We wish to highlight that the discussion of NK cells in NPC was addressed in a separate section (section 6).

Perhaps due to difficulties in establishing EBV-infected non-transformed epithelial cell model, there is lack of literature addressing how NK cell restrict EBV infection in epithelial cells. The closest is the evaluation of NK against NPC cell lines, but this does not address how the NK restrict EBV in epithelial cells in the normal physiological context. This disparity is highlighted in section 8, paragraph 1.

  1. Section 5: It would be more logical to add a short paragraph at the beginning of this section reminding readers that Type II latency program is associated with lower BZL1 expression, higher miR-BARTs, and higher LMPs and each of these could have a role in NK evasion.

>> We thank the reviewer for the suggestion. We have included an introductory paragraph in section 5 as recommended for better clarity.

  1. It would be also interesting to discuss therapeutics to enhance NK cells and their potential benefit to NPCs (e.g PMID: 26000725)

>> We thank the reviewer for the suggestion. We have created a new section 8 “Translational applications of NK cells in NPC” to address this.
